# A Comparison of Etiology, Pathogenesis, Vaccinal and Antiviral Drug Development between Influenza and COVID-19

**DOI:** 10.3390/ijms24076369

**Published:** 2023-03-28

**Authors:** Junhao Luo, Zhuohan Zhang, Song Zhao, Rongbao Gao

**Affiliations:** 1NHC Key Laboratory of Biosafety, National Institute for Viral Disease Control and Prevention, Chinese Center for Disease Control and Prevention, Beijing 102206, Chinazzhdkyyx@163.com (Z.Z.);; 2NHC Key Laboratory of Medical Virology and Viral Diseases, National Institute for Viral Disease Control and Prevention, Chinese Center for Disease Control and Prevention, Beijing 102206, China; 3Chinese National Influenza Center, National Institute for Viral Disease Control and Prevention, Chinese Center for Disease Control and Prevention, Beijing 102206, China

**Keywords:** SARS-CoV-2, influenza virus, etiology, spread, pathogenesis, vaccines, antiviral drug

## Abstract

Influenza virus and coronavirus, two kinds of pathogens that exist widely in nature, are common emerging pathogens that cause respiratory tract infections in humans. In December 2019, a novel coronavirus SARS-CoV-2 emerged, causing a severe respiratory infection named COVID-19 in humans, and raising a global pandemic which has persisted in the world for almost three years. Influenza virus, a seasonally circulating respiratory pathogen, has caused four global pandemics in humans since 1918 by the emergence of novel variants. Studies have shown that there are certain similarities in transmission mode and pathogenesis between influenza and COVID-19, and vaccination and antiviral drugs are considered to have positive roles as well as several limitations in the prevention and control of both diseases. Comparative understandings would be helpful to the prevention and control of these diseases. Here, we review the study progress in the etiology, pathogenesis, vaccine and antiviral drug development for the two diseases.

## 1. Introduction

Influenza viruses and coronaviruses are two common pathogens of novel respiratory viruses to have emerged in humans. According to previous reports, at least 17 novel respiratory viruses have been reported to cause human infection worldwide since the 1990s, eight and five of which respectively belong to influenza viruses (H5N1, H9N2, H7N7, H1N1 swine influenza virus, H3N2 swine influenza virus, H1N1pdm09, H7N9 and H5N6) and coronaviruses (SARS-CoV, CoV-NL63, CoV-HKU1, MERS-CoV and SARS-CoV-2) [1]. Among these emerged viruses, both H1N1pdm09 and SARS-CoV-2 have caused global pandemics. H1N1pdm09 was first detected in the spring of 2009, and resulted in a pandemic as declared by WHO around three months later. SAR-CoV-2 was first detected at the end of 2019, and resulted in a pandemic as declared by WHO around three months later. SAR-CoV-2 and influenza viruses share many similarities in terms of pathogenic transmission and pathogenic properties, although they belong to different virus families. Vaccines and antiviral drugs have shown some definite efficacy in the prevention and control of both diseases, however, both also have many unsatisfactory aspects in their efficacy. Here, to enhance the understanding of the two viruses or diseases, we made a comparative review of the research progress in their etiology, pathogenesis, vaccine and antiviral drug development.

## 2. Etiological Characteristics

Both influenza virus and SARS-CoV-2 are RNA viruses that cause respiratory tract infection by respiratory droplet transmission. The two viruses present differently in their genomic characteristics, size, and encoding protein or receptor binding properties, whereas they have many similarities in terms of viral morphology, infection route, transmission mode and transmission ability, as detailed in Table 1.

### 2.1. Evolutionary Characteristics of the Genome

#### 2.1.1. Influenza Virus

Influenza virus, a member of Orthomyxoviridae, exists widely in nature. There are four types of influenza viruses: influenza A, B, C, and D; however, only influenza A and B viruses cause clinically important human disease and seasonal epidemics. Influenza A virus is the most complex and pathogenic type, and is divided into several subtypes (HxNy) according to the antigenicity and combination of its surface protein hemagglutinin (HA) and neuraminidase (NA), of which there are 18 HA (H1–H18) subtypes and 11 NA (N1–N11) subtypes. They can infect humans and many other animals including birds, pigs, horses, and bats. The caused diseases are known as human influenza, avian influenza, swine influenza, and equine influenza, because these viruses are species-specific in causing infection. Novel influenza viruses, which emerge frequently especially in influenza A viruses due to mutation and reassortment, may result in prevalence and even pandemics, which has been considered to be inevitable at an uncertain point in the future.

Studies on genomic evolution have shown that the evolutionary route of the influenza virus is closely related to big flu epidemics in history, especially to the four global pandemics of 1918, 1957, 1968 and 2009, which had the most profound impact on the evolution of the influenza virus and on the formation of seasonal influenza [13]. Studies have now found that the 1918 pandemic H1N1 virus originated from an avian source, adapted and spread in humans after cross-species infection, and thereafter became a seasonal virus. The 1957 pandemic H2N2 was a novel recombinant virus which was formed after replacing the HA, NA and PB1 genes of the human H1N1 virus by those of avian H2N2. The 1968 pandemic H3N2 virus was also a novel recombinant virus which was formed by replacing the HA and PB1 genes of the human H2N2 virus by those of avian H3. H1N1 and H3N2 cocirculated and became seasonal epidemics in humans after H1N1 influenza re-emerged in 1977. The currently prevalent seasonal H1N1 virus was the 2009 pandemic H1N1 virus, which was a novel quadruple reassortant influenza virus, of which the genes PB2 and PA were derived from avian influenza viruses of North American lineages, the gene PB1 was derived from human seasonal influenza (H3N2) viruses, the genes HA, NP, and NS originated from classical swine influenza A viruses of North American lineages, and the genes NA and M originated from swine influenza virus of Eurasian lineage [14,15,16].

#### 2.1.2. SARS-CoV-2

Coronavirus, another large virus family widely existing in nature, is identified to infect only vertebrates, such as humans, pigs, cats, dogs, wolves, chickens, cattle, poultry, etc. SARS-CoV-2 is the seventh coronavirus known to infect humans besides HCoV-229E, HCoV-OC43, HCoV-NL63, HCoV-HKU1, SARS-CoV and MERS-CoV. Among them, SARS-CoV, MERS-CoV and SARS-CoV-2 can all cause severe respiratory syndrome manifestations in humans [17,18,19,20,21,22]. The comparative genetic analysis showed that SARS-CoV-2 and SARS-CoV have 79.6% identity in genomic sequences and share 76.5% homology in S protein, and that SARS-CoV-2 and MERS-CoV have 51.8% identity in genomic sequences and share 28.9% homology in S protein [8,23]. The traceability study showed that SARS-CoV-2 and the bat coronavirus (Beta CoV/bat/Yunnan/RaTG13/2013) detected in the chrysanthemum-headed bat have 96.2% identity in genomic sequences and share 97% homology in S protein, concluding that SARS-CoV-2 may derive from a variant of bat coronavirus [24]. Mutations with high-frequency substitution (C > U) have been observed in the two viruses, suggesting that SARS-CoV-2 has undergone a considerable period of evolution [25]. With the worldwide prevalence of SARS-CoV-2, the virus has been undergoing continuous changes, and a variety of viral variants has gradually been formed with enhanced transmissibility but attenuated pathogenicity or increased immune escape level, among which the following six variants were considered to be representative variants of concern (Table 2) [26]. Accumulating evidence has indicated that SARS-CoV-2 will coexist with human beings and may eventually become a common respiratory infectious agent presenting seasonal epidemics [27,28].

#### 2.1.3. Mechanisms of Mutation

The mutation mechanisms of RNA viruses include point mutation, reassortment and recombination. Point mutation occurs most commonly because the genome of RNA viruses is prone to errors during replication. Point mutation generally leads to antigenic drift, creating new virus subtypes that alter antigenic structure, pathogenicity or drug resistance, for example, the 627-K PB2 mutation of human seasonal influenza virus strains, human pathogenic H5N1, H7N9, H10N8 viruses [29,30], D614G mutation of SARS-CoV-2, etc. [31]. Antigenic drift caused by accumulative point mutation, a main type of antigenicity change in the influenza virus, may lead to localized outbreaks or big endemics of influenza virus, since a virus with antigenic drift over a period of time becomes very different from the initially circulating virus. A special type of mutation for influenza is reassortment, the major reason for a pandemic to arise. The segmented genome of influenza viruses allows for the exchange of RNA fragments between genotypically distinct influenza viruses, resulting in new strains and/or subtypes, which is known as reassortment. Hence, reassortment generally results in big mutations of viruses.

Genetic recombination is one of the main processes that produce genetic diversity acted upon by natural selection. As for viruses, recombination within different lineages requires the co-circulation and the co-infection of the viruses in the same host. Recombination between divergent variants has generated new genotypes and may have played a crucial role in the evolution of H5N1 viruses [32]. For SARS-CoV-2, the most recent recombinant is the lineage labeled as XBB. XBB lineage is a recombinant of BJ.1 and BM.1.1.1, both belonging to the BA.2 lineage [33].

### 2.2. Pathogenic Characteristics

#### 2.2.1. Human Infection with Seasonal Influenza

Influenza epidemics in different regions show different characteristics, because the prevalence of seasonal influenza viruses is affected by various factors including temperature, sunshine hours, and maximum rainfall. Seasonal influenza epidemics occur mainly in the winter in temperate regions, while irregular epidemics are present throughout the year in tropical regions. Seasonal influenza epidemics, in addition to causing large numbers of population infections each year, can cause plenty of deaths globally despite a relatively low fatality rate. A 2019 study led by the US CDC estimated that global seasonal influenza infection-related deaths reach 291,243–645,832 each year, through estimations of country-specific influenza-associated respiratory excess mortality rates for 33 countries using time series log-linear regression models with vital death records and influenza surveillance data [34]. Thus, seasonal influenza has a significant and ongoing impact on human health.

The incubation period for seasonal influenza virus infection is generally 1–7 days, mostly 2–4 days. If there are no complications, the process is mostly self-limiting, and the symptom of high fever gradually recovers on day 3–4 after the illness onset. In addition, about one third of cases of seasonal influenza are asymptomatic. The initial symptom of seasonal influenza may be characterized by the onset of sudden illness with nonspecific symptoms such as fever, chills, headache, muscle pain, and loss of appetite, followed by respiratory symptoms such as dry cough, sore throat or dry throat, nasal congestion, and runny nose. These initial presentations may be less dramatic in the elderly and in those with compromised immune systems, however, it is relatively frequent to progress to severe lower respiratory disease after experiencing initially mild symptoms including fever or malaise alone without typical respiratory symptoms. Some children may present with fever alone without other observable symptoms. However, some cases with seasonal influenza infection may develop into severe disease which may be complicated by cardiac damage, neurological damage, and even multi-organ failure or diffuse intravascular coagulation in critically ill patients.

#### 2.2.2. Human Infection with Pandemic Influenza

Influenza A viruses have caused four pandemics in the past hundred years: H1N1 Spanish influenza in 1918, H2N2 Asian influenza in 1957, H3N2 Hong Kong influenza in 1968, and H1N1 swine influenza in 2009, and these viruses became prevalent seasonal strains in subsequent years. During each pandemic, a novel influenza virus arose, either directly from an avian host (1918), or via reassortment between an avian virus and a circulating human strain (1957 and 1968), or through influenza virus reassortment in pigs (2009), and spread through the human population [35]. Pre-existing antibodies against the influenza virus in the human body cannot provide efficient protection to emerging pandemic viruses, hence the population is generally susceptible to these viruses, thus causing widespread epidemics.

Each influenza pandemic has resulted in a large number of deaths of infected people, ranging from hundreds of thousands (2009 H1N1 pandemic) to tens of millions (1918 H1N1 pandemic). The disease manifestations caused by the pandemic influenza virus have diverse presentations, ranging from asymptomatic, mild upper respiratory tract infections to severe or fatal pneumonia with a series of severe complications including acute respiratory distress syndrome and multiple organ failure. The 2009 novel influenza A (H1N1) pandemic, the mildest influenza pandemic in history, still caused 151,700 to 575,400 deaths worldwide during the pandemic. The clinical manifestations of infected cases were similar to those of seasonal influenza. Although most cases presented self-limiting symptoms, including sudden fever, fatigue and anorexia, and were followed by upper respiratory symptoms such as cough, sore throat and nasal congestion, some cases with the 2009 pandemic influenza developed severe pneumonia, acute respiratory distress syndrome (ARDS), pulmonary hemorrhage, pleural effusion, renal failure, sepsis, shock and Reye syndrome, respiratory failure and multiple organ damage, which can lead to death [36].

#### 2.2.3. Human Infection with Animal Influenza

Although cross-species transmission of influenza viruses do not readily occur due to their strict species-specificity, avian and swine influenza viruses are still of great public health concern. Genetic and molecular evidence has shown that the past four influenza pandemics originated from or were related to avian or swine influenza viruses, and many sporadic cases of human infection with animal influenza viruses have occurred from time to time since the 20th century, either by direct cross-species infection or by reassortment of viruses from multiple genera. Therefore, avian and swine influenza viruses are of major concern in the field of influenza pandemic preparedness.

Animal influenza viruses that have been reported to infect humans include H5N1, H9N2, H7N9, H7N2, H7N3, H7N4, H10N8, H10N3, H6N1 and H5N6 avian influenza viruses, and H1N1, H1N2 and H3N2 swine influenza viruses, of which H7N9, H5N1 and H5N6 caused the top three cases and resulted in high fatality. Since the first human case of H7N9 infection was identified in China in 2013, as of September 2021, a total of 1568 cases with H7N9 infection have been reported globally, involving three countries, with a case fatality rate of 39.2% [37]. After an outbreak of A(H5N1) virus in 1997 in poultry in Hong Kong SAR, China, since 2003, the viruses have spread from Asia to Europe and Africa with some spillover to humans; by 5 January 2023, a total of 868 human infections with 457 deaths had been reported in 21 countries globally [38]. By the end of 2021, a total of 66 human cases of H5N6 infection with 36 deaths have been reported worldwide since the first case of H5N6 infection was identified in Sichuan, China in 2014. Of those, 65 H5N6 cases were in China, and the remaining case was reported in Laos [39]. The clinical features of the three viral infections are similar to those of severe pandemic cases, with sudden high fever, headache, muscle soreness and rapidly progressive bilateral pneumonia, which could be accompanied by hemoptysis, ARDS, septic shock or/and multiple organ failure [40,41,42,43,44,45,46,47,48]. Infections with the other animal influenza viruses presented diverse symptoms including mild respiratory symptoms or asymptomatic infection in most cases, and conjunctivitis in some cases with H7 infection.

#### 2.2.4. Human Infection with SARS-CoV-2

COVID-19 caused by SARS-CoV-2, as with influenza, may present a spectrum of clinical manifestations, from asymptomatic to critical illness. The majority of symptomatic patients usually present with fever, cough and shortness of breath, and some cases have sore throat, hyposmia, anorexia, nausea, malaise, muscle atrophy and diarrhea [49,50,51,52]. Stokes et al. reported that among 373,883 symptomatic COVID-19 cases diagnosed in the United States, 70% presented with fever, cough, and shortness of breath, 36% presented with muscle pain, and 34% presented with headache [53]. A report by the Chinese Center for Disease Control and Prevention showed that 81% of patients presented with mild, 14% with severe (including tachypnea, hypoxia, or imaging abnormalities), and 5% with critical disease (including respiratory failure, shock, multiple organ dysfunction), and the average fatality was 2.3% in 2020 [52]. Another large meta-analysis reported laboratory abnormalities including lymphopenia (47.6%), elevated C-reactive protein levels (65.9%), elevated cardiac enzymes (49.4%), abnormal liver function (26.4%), and abnormal renal function (10.9%) in 8697 COVID-19 patients in China [54]. However, with the emergence of the Omicron strain, the pathogenicity and lethality of SARS-CoV-2 were significantly decreased [55], and the vast majority of cases only presented with acute influenza-like upper respiratory tract infection symptoms, such as nasal congestion, cough, expectoration, and sore throat [56,57], however, losses or alterations of the sense of smell or taste were more common than in influenza infection [58]. Additionally, SARS-CoV-2 is a pathogen which can definitely cause zoonosis. However, the intermediate animal host, a very important concern for emerging infectious disease, has not yet been identified. Although several animals such as mink [59] and pangolin [60] have been shown to be infected by SARS-CoV-2 naturally, there is no clear evidence to support the transmission of SARS-CoV-2 from a definite animal to humans.

#### 2.2.5. Co-Infection of SARS-CoV-2 and Influenza Virus

Several investigations have demonstrated that co-infections with SARS-CoV-2 and influenza virus were frequent but not common in clinics. Zhu et al. reported that among 257 patients with COVID-19, 0.8% were co-infected with influenza A. Although SARS-CoV-2 mimics the influenza virus regarding clinical presentation, transmission mechanism, and seasonal coincidence, co-infection with influenza viruses is uncommon in patients with SARS-CoV-2 infection. However, during the high season of seasonal influenza, the risk of co-infection may spike in COVID-19 patients [61].

Studies suggest that co-infection can increase the severity of symptoms of both diseases. COVID-19 patients with influenza co-infection had higher mean hospitalization costs and total lengths of stay, higher odds of needing mechanical ventilation, and higher in-hospital mortality relative to the COVID-positive and influenza-negative cohort [62]. However, a protective effect of co-infection with the influenza virus was indicated in patients with COVID-19 in an investigation [63]. Hence, more investigations on co-infection with SARS-CoV-2 and influenza virus would be helpful to understand the associated problems.

### 2.3. Pathogenesis

For all hosts facing infection, a fundamental challenge is how to balance between clearance of the pathogen and maintenance of tissue function. A robust immune response may rapidly clear pathogens, but may also cause extensive collateral damage of tissue function. Achieving this balance is particularly important in the respiratory tract because the host cannot survive without adequate gas exchange in the lungs. In turn, hosts adopt different defense strategies, depending on the type of pathogen, the timeliness of infection, the affected tissue, etc. Both the virus itself and the host immune response are involved in the pathogenesis of the influenza virus and SARS-CoV-2.

#### 2.3.1. Pathogenesis of the Influenza Virus

Influenza viruses infect the epithelial cells of the upper/lower respiratory tract by binding viral HA protein to SAα2,6-Gal/SAα2,3-Gal linked receptors. Seasonal and pandemic strains are specific for SAα2,6-Gal linked receptors, which are predominantly expressed in the human trachea, whereas the avian influenza virus preferentially binds to SAα2,3-Gal linked receptors, which are predominantly expressed in alveolar type II cells. In early seasonal influenza infection, microscopic examination of tracheal and bronchial biopsy specimens revealed diffuse epithelial exfoliation, infiltration of monocytes as the predominant inflammatory cells, bloody exudates filling in the airway lumen with interstitial swelling, and formation of small-vessel thrombosis in the bronchiole walls. In severe cases, alveolar tissue was injured resulting in similar epithelial necrosis and even collapse to the proximal site. In mild cases of pandemic influenza, the characteristics of tissue damage were similar to those of seasonal influenza, while severe cases mainly showed parenchymal lung tissue damage accompanied by pathological changes in other organs. Autopsy studies showed that the pathological features were pulmonary edema, diffuse alveolar damage (DAD), small pulmonary vascular thrombosis and combined bacterial pneumonia in fatal cases with 2009 influenza A (H1N1) virus infection; and cerebral edema, hepatic congestion, and adrenal hemorrhagic necrosis in some fatal cases with 2009 influenza A (H1N1) virus infection. Immunohistochemical staining and electron microscopy showed that the virus was mainly distributed in lung epithelial cells, with less distribution in submucosal glands, alveolar macrophages and endothelial cells, but not detected in non-respiratory tissues [64,65].

There are several widely accepted pathogenic mechanisms of influenza viruses including: ① Influenza viruses induce apoptosis of host cells. A large number of daughter viruses are produced through a series of biochemical processes of replication, and then spread to infect other host cells after influenza viruses invade the host cells, resulting in diffusely cytopathic changes including apoptosis. Influenza viruses induce apoptosis in a variety of cell types both in vitro and in vivo through Fas/FasL mediated mechanism, NF-κβ-mediated pathway promoted by IFN-α/βin infected cells, signal transduction molecules and P38 mitogen-activated protein kinase regulatory pathway, or through direct stimulation of viral proteins (NS1, PB1-F2) [66]. ② Oxidative stress, an imbalanced state with oxidation tendency between oxidative and antioxidant effects in vivo, results in infiltration of inflammatory neutrophils, increased secretion of protease, production of a large number of reactive oxygen species (ROS), and finally, injury to infected cells. Excess ROS induced by influenza virus may trigger oxidative damage to tissue cells, while the antioxidant system in the body is unable to counteract the oxidative stress effect, and the damage may not be repaired in a timely manner [67]. Influenza A virus infection increases plasma and urine levels of metabolites such as 8-hydroxydeoxyguanosine, malondialdehyde, 2-isoprostanes, 7-ketocholesterol, 7-β-hydroxycholesterol, and carbonyl compounds. In patients with H1N1 infection, levels of antioxidant enzymes such as SOD and catalase; cytokines such as IL-6, 1L10 and TNF-α; and HSPs such as HSP90 and HSP27 were also elevated [68]. ③ Cytokine storm, a phenomenon that occurs more frequently in some severe influenza cases, especially in severe cases of pandemic influenza and avian influenza, may cause extensive lung tissue edema, pneumonia, alveolar hemorrhage, and even promote the formation of ARDS or multiple organ failure. Cytokines are produced rapidly by epithelial cells and immune cells of the respiratory mucosa after influenza infection, and activate immune cells to trigger local and systemic immune response, to promote the production of a series of cytokines that play a corresponding regulatory role in the immune process against viral infection [69,70]. However, cytokine is a double-edged sword which contributes to immune regulation against influenza virus as well as to severe immunopathological damage to the host if overexpression of cytokines occurs.

However, the repair mechanisms of the extensively damaged airway or alveolar epithelium caused by the influenza virus are not yet fully understood. Influenza virus infection results in large areas of basement membrane exfoliation, loss of lung microstructure, and the formation of partial or complete pulmonary atelectasis in the upper/lower airway. Nonetheless, the damaged or denervated epitheliums are replaced by the proliferation of one or more airway or alveolar progenitor cells. Hence, a strong regenerative response, including termination of inflammation, stromal deposition, proliferation of progenitor cells and/or reconstruction of the alveolar capillary barrier, is necessary to restore gas exchange and to protect the host from secondary microbial infections [71].

#### 2.3.2. Pathogenesis of COVID-19

Autopsy testing of COVID-19 fatal cases showed that lung tissue was the leading damage, of which DAD was frequently observed. The damage presented typical features, including reactive type II pulmonary cell and interstitial edema, bronchial and bronchiolar inflammation, etc. SARS-CoV-2 proteins were detected in both alveolar epithelia and ciliated epithelia, as well as in submucosal glands and lymphocytes. In addition to respiratory histiocytes, viral proteins or ultrastructure were detected or observed in renal epithelia and intestinal cells in some case, whereas viral RNAs were detected in more multiple extrapulmonary tissues including the liver, spleen, brain and even heart of autopsy cases, although the viral load in lung tissue was the highest [72,73]. These data indicate that, in addition to respiratory tissues, the direct damage caused by SARS-CoV-2, which can spread to multiple extrapulmonary organs of the host, played a role in damage to the respiratory tract and possibly in injuries to extrapulmonary tissues. Studies have shown a positive correlation between the severity of cases and the level of viral load in COVID-19 [8,74,75,76].

In addition to the direct damage of the virus, the excessive immune response of the host plays an important role in the pathogenesis of COVID-19. Viral infection and tissue damage trigger local immune responses including recruiting macrophages and monocytes, releasing cytokines, and initiating adaptive T and B cell immune responses. In severe patients, the infection may cause dysfunctional immune responses, trigger a cytokine storm, and induce immunopathological damage, mediating extensive inflammation in the lungs, and resulting in severe pulmonary and even systemic pathological injuries which are an important cause of ARDS and multiple organ failure. Studies have shown that elevated IL-6 levels positively correlate with the severity of COVID-19; IP-10, MCP-3 and IL-1RA levels were elevated in severe patients and correlated with disease progression and severity, and plasma IP-10 levels were significantly correlated with plasma viral load [77,78,79,80]. Oxidative stress in SARS-CoV-2 infection (especially in severe cases) also exhibits damaging effects on the organism similar to those seen in influenza virus infection. A study by Nairrita et al. [81]. demonstrated significant redox imbalance in subjects with COVID-19. There were significant differences in serum NOx levels between non-ICU and ICU admitted patients, and a positive association between the serum levels of VCAM-1 and ICAM-1; and a negative association between ascorbate radical and mortality in COVID subjects was observed, while IL-17c and TSLP levels predicted the need for intensive care in COVID-19 subjects.

In addition, COVID-19 patients may present unique pathological changes in pulmonary vessels, including thrombosis and endothelial damage, which may be related to the SARS-CoV-2 binding receptor ACE2. SARS-CoV-2 infection decreases the function of ACE-2 which can regulate the renin–angiotensin system, and may then result in vascular endothelial damage with increased permeability, and an inflammatory cascade effect because of dysfunction of the renin–angiotensin system, and may furtherly result in pulmonary edema, abnormal coagulation or disseminated intravascular coagulation [82,83,84].

## 3. Vaccines

### 3.1. Influenza Virus Vaccines

Since the 1930s, several types of influenza vaccines have been developed, including inactivated split influenza vaccine, subunit vaccine, recombinant protein vaccine, and live attenuated influenza vaccine [85]. The characteristics and scope of use of these different vaccines are listed in Table 3. At present, globally, the most vaccinations of influenza vaccine are still the inactivated split vaccine with whole virus which is propagated in 9–11-day old chicken embryos. However, successive passages of influenza viruses, especially the H3N2 subtype, can lead to antigenic changes at antigenic sites due to adaptive mutations in chicken embryos, thus reducing vaccine efficacy [86,87]. In order to overcome this shortcoming, cell culture-based vaccines were developed and approved for use in Europe in 2007, and in 2011 for MDCK and Vero cell culture-based influenza virus vaccines, and in 2013 for recombinant insect cell culture-based influenza virus vaccines in the United States [85].

The influenza virus vaccine candidate strains are usually confirmed or updated in March and September each year by the WHO, based on etiological data from the global influenza surveillance network in the northern and southern hemispheres, respectively. The updated seed strains are sent to the approved vaccine industries globally by WHO influenza vaccine reference laboratories. Therefore, for upcoming or ongoing influenza epidemics, there is a lag of about six months or longer because of the production cycle of vaccines, whose efficacy would be greatly impacted if an unexpected influenza epidemic was caused by a variant [88,89]. Currently, the trivalent influenza vaccine (TIV) and the quadrivalent influenza vaccine (QIV) are available for vaccination in most countries. The TIV component contains subtype A (H3N2), subtype A (H1N1) pdm09, and one lineage of the B strain, whereas the QIV component contains two lineages of the B strain. Theoretically, TIV has, on average, a 25% chance of mismatching with the current year’s prevalent strain because of lack of one lineage of type B strains [90], whereas the QIV vaccine makes up for this deficiency [91].

However, influenza vaccination cannot induce a long-lasting immunity. The maintenance of the efficiently protective antibody induced by the influenza vaccine is usually 6–12 months in vivo. Thus, annual vaccination is recommended in high-risk groups. The efficacy of the influenza vaccine fluctuates between 10% and 83% depending on the match between the seed strain and the prevalent strain of the year [92,93]. In addition, vaccine effectiveness is greatly reduced or absent in older adults, children, and people with underlying diseases [92,94]. On the whole, the actual effectiveness of the influenza vaccine is far from people’s expectations. In addition, due to the limitation of the efficacy and the lack of cross-protection against other subtypes, the available vaccines may not provide effective protection when new subtypes or new variants become prevalent. The development of a universal influenza vaccine has become the desired goal in the field; several universal influenza vaccines are currently in clinical trials. Universal influenza vaccine candidate H8/1 and H5/1, based on HA studies at the Icahn School of Medicine at Mount Sinai, is in Phase I clinical trial [95]. Universal influenza vaccine candidate Multimeric-001 (M-001), developed by Biond Vax of Israel in 2020, is in Phase III clinical trials [96]. Another universal influenza vaccine candidate FLU-v, that has been shown to have good population protection and to reduce the incidence of moderate influenza disease in clinical trials, is a synthetic peptide vaccine developed by Pep T cell company in the United Kingdom [97,98].
ijms-24-06369-t003_Table 3Table 3Influenza Vaccine Types and Characteristics.Type of VaccineAdvantagesDisadvantagesRecommended PopulationEfficacy of ImmunizationInactivated virus vaccineHigh safety; Contains intact viral antigenic components; High immunogenicity; No risk of restoring virulence [99,100].Low cross-protection;Relatively low antibody production rate; Relatively low efficacy in inducing cellular immune response; Relatively high side effect of fever [101,102].Recommended for people aged six months or older [103].Variable in different years depending the matching degree of vaccine stains to circulating strains. Pooled efficacy 59% (95% CI: 51–67%) in adults aged 18–65 years [104], 44–65% in pregnant women [105,106], 54% (95% CI: 37–66%) in babies aged 6–35 months [107], 55% (95% CI: 46–62%) in children aged 2–18 years [108], 58%(95% CI: 34–73%) in people aged 65 or older [109].Inactivated split virus vaccineReducing unwanted side effects that may be caused by other components of the virion [110,111].Potential allergic reaction to eggs.Recommended for people aged six months or older [112].No available data.Subunit vaccineHigh safety; Rapid; Stable; Scalable production; Induction of humoral and cellular immune responses; Potential cross-protection [113].Relatively low immunogenicity; Relatively high manufacturing costs [112]; Appropriate adjuvants, suitable routes of administration [113].Healthy children, adults and the elderly [112].No available data.Live attenuated vaccineBetter cellular immune responses induced by LAIV in infants and school-age children than in adults [114,115,116,117]; Eliciting broadly cross-reactive T cells which are not induced by traditional inactivated influenza vaccines [115].Risk of restoring virulence [118,119]; Potential allergic reaction to eggs.Better in infants and school-age children than in adults [114,116,117].46% (95% CI: 7–69%) in children aged 2–17 years in the 2015–2016 season in USA [114].


### 3.2. SARS-CoV-2 Vaccines

According to WHO statistics, as of 2nd December 2022, a total of 175 SARS-CoV-2 vaccines under development had entered clinical trials around the world, and over fifty SARS-CoV-2 vaccines had been approved for conditional marketing or emergency use worldwide, including adenovirus vector-based vaccines, inactivated vaccines, mRNA vaccines, and subunit vaccines. The characteristics of different types of vaccines are listed in Table 4. Among them, the mRNA vaccine is the first nucleic acid vaccine to be vaccinated for the administration of infectious diseases, and may activate both the CD4+ T cell exogenous immune response and the CD8+ T cell endogenous immune response pathways to induce the production of abundant effector and memory CD8+ T cells [120]. Compared with the protein subunit vaccine, the mRNA vaccine may be faster and cheaper to develop. However, a major obstacle is the degradability of mRNA on application of the mRNA vaccine. To overcome this obstacle, a strategy of encapsulating mRNA in lipid nanoparticles, which also function as adjuvants, has been used to stabilize the vaccine structure [121]. In contrast, the inactivated virus vaccine is one of the most successful classical vaccines historically, and has been proven to be highly stable and viable for storage and transport [122]. In addition, the development cycle of inactivated vaccines is relatively short compared with other vaccine types, making it the first type of COVID-19 vaccine to be marketed worldwide. However, its long manufacturing time or requirement of a BSL-3 facility may put it at a disadvantage in the current COVID-19 pandemic [123,124].

With the prevalence of SARS-CoV-2 and the emergence of novel variants, the effectiveness of vaccines in preventing infection has shown a marked gradual decline. A recent meta-analysis showed that, on average, one to six months after two-dose vaccination, vaccine effectiveness against SARS-CoV-2 infection decreased by 21% (95% CI 13.9–29.8%) in all age groups and by 20.7% (95% CI 10.2–36.6) in the elderly, and decreased by 24.9% (95% CI 13.4–41.6%) for the prevention of symptomatic infection in all age groups and by 32.0% (95% CI 11.0–69.0%) in the elderly. However, 81% of vaccines in the 78 studies analyzed still showed greater than 70% efficacy in preventing severe disease [139]. Meanwhile, the frequency of vaccination has presented an increasing trend, from the initial two doses to an additional one-dose booster or even additional two-dose booster in most countries [140]. Furthermore, the vaccination mode has changed from vaccination with a single type of vaccine to sequential vaccination with different types of vaccines [141]. In addition, the components of the vaccine changed from monovalent vaccine with the α strain, bivalent vaccine containing the α and delta strains, trivalent vaccine containing the additional Omicron strain, to tetravalent vaccine containing the additional ß strain [142,143]. In the future, will the SARS-CoV-2 vaccine need updated components annually, or be recommended as an annual vaccination for high-risk populations, as with the influenza vaccine? This is yet unknown, but may be a possibility.

## 4. Antiviral Drugs

### 4.1. Anti-Influenza Virus Drugs

Influenza viruses bind the sialic acid (SA) receptor to adsorb to the cell surface by their surface HA, and then enter the cell to form infection through host-mediated endocytosis. Under low pH conditions, the HA fusion conformation is altered to promote virus fusion with endosomal membranes. Simultaneously, the non-glycosylated matrix protein M2 ion channel is activated, and inward protons enter the viral inner membrane, which causes the dissociation of matrix protein 1 (M1) from viral ribonucleoprotein (vRNP). Then, the dissociated vRNP is subsequently released into the cytoplasm and migrates to the nucleus where viral components are transcribed or replicated. The basic proteins PB1and PB2, and the acidic protein PA and NP are necessary for the transcription and replication of the influenza virus. The newly formed daughter viruses vRNP and structural proteins are exported to the cytoplasm for packaging and maturation with the help of M1 and non-structural protein 2 (NS2). Daughter viruses are budded and released from infected cells after NA removes sialic acid residues on the membranes of infected cells and viral membranes. Inhibition of the functional molecules associated with replication may inhibit virus replication because each stage of influenza virus replication requires the coordination and participation of its own proteins and host proteins (Figure 1).

At present, anti-influenza virus drugs can be divided into two categories: The first is inhibitors targeting structural or functional proteins of the influenza virus, including neuraminidase inhibitors, M2 ion channel blockers, RNA polymerase inhibitors, hemagglutinin inhibitors, etc. The second is drugs that act on the infected host. The replication of influenza virus is highly dependent on host cells, therefore, in order to effectively deal with the drug resistance caused by the high mutation rate of the virus, researchers are looking for potential targets of the host that can be used as influenza therapeutic drugs including SA inhibitors, protease inhibitors, V-ATPase, and host signaling pathway inhibitors, etc. The characteristics of the different drugs are detailed in Table 5.

Among these drugs, the M2 ion channel blockers Amantadine and Rimantadine have shown severe resistance to the currently prevalent strains and are no longer recommended for use in clinics. Currently, the main drugs in clinical use for specific anti-influenza virus include the neuraminidase inhibitors including Oseltamivir and Peramivir, the RNA polymerase inhibitors including Baloxavir marboxil, marketed in different countries [157], and Favipiravir [158] and Arbidol [158] with a broad-spectrum antiviral effect. These drugs have shown some efficiency in shortening the course of the disease and in reducing the risk of severe disease if used early [159,160,161], and are usually recommended for use within 48 h after the onset of illness [161,162]. However, studies have demonstrated that the efficiency of antiviral drugs is greatly reduced once the host cascade immune responses are stimulated by the infective virus [161,163]. In addition, viral strains with reduced susceptibility or resistance to them are emerging, and therefore, drug susceptibility monitoring for prevalent strains has become one of the main concerns for influenza virus surveillance since the use of these drugs increased.

### 4.2. Drugs against SARS-CoV-2

The single-stranded RNA in the SARS-CoV-2 viral genome is translated by the host mechanism to produce viral peptides pp 1a and pp 1ab, which are processed to synthesize Nsps after undergoing proteolytic cleavage by PLpro and 3CLpro proteins. These Nsps encode the replication transcription complex (RTC). RTC produces a series of sub-genomic messenger RNAs encoding accessory and structural proteins after continuously replicating. Daughter SARS-CoV-2 particles are assembled with viral genomic RNA and proteins in the ER–Golgi-intermediated compartment. The virus particles are released from infected cells after vesicle-containing viruses fuse with the plasma membrane of infected cells. SARS-CoV-2 may trigger human innate immune responses including inducing interferon (IFN) after infection. However, upregulation of IFN levels activates the expression of the ACE2 receptor of SARS-CoV-2 because ACE2 is also a human interferon-stimulating gene. The interaction between the viral S protein and ACE2 results in the cleavage and activation of the Furin-mediated S1–S2 boundary and transmembrane protease serine 2 (TMPRSS2)-mediated S2 site [164]. Then, the S2 subunit induces fusion of the virus and host cell plasma membranes and release of the viral nucleic acid. Lung type II pneumocytes and nasal goblet secretory cells with co-expression of ACE2 and TMPRSS2 are the main target cells of SARS-CoV-2 invasion in vivo [165]. The effective anti-SARS-CoV-2 drugs that have been studied in vitro and in vivo can also be divided into two categories (Figure 2): One is those targeting the various protein molecules that play a role in the invasion of SARS-CoV-2 into host cells, including monoclonal antibody drugs (there are at least 23 monoclonal antibodies in clinical trials [166], including bamlanivimab, etesivimab, casirivimab, imdevimab, adintrevimab, regdavimab, sotrovimab, tixagevimab, cilgavimab, Ronapreve, and Evusheld), membrane fusion inhibitors (including Arbidol which has entered clinical trials [167,168], and EK1 and EK1C4 which are entering preclinical evaluation [169]), TMPRSS2 inhibitors (including, MI-432, MI-1900 [170], and a variety of Camostat analogs [171]), furin inhibitors (including decanoyl-RVKR-chloromethyl ketone and naphthofluorescein [172]), and drugs targeting ACE2 (including RhACE2 APN01, which has entered phase II clinical trials [173], chloroquine and hydroxychloroquine [174]). The other category is those targeting key molecules associated with SARS-CoV-2 replication in the host, including RNA-dependent RNA polymerase (RdRP) inhibitors (including Favipiravir [175], Molnupiravir [176]), peptidomimetic 3CLpro inhibitors (including PAXLOVIDTM [177]), and PLpro inhibitors (including GRL0617 [178], VIR250, VIR251 [179] and acriflavine [180]).

However, as clinical trials expand and viral variants emerge, the efficacy of many drugs presents big changes. For example, most antibodies present significantly decreased or even absent efficacy [181]. Inhibitors such as Arbidol [182], chloroquine and hydroxychloroquine [183] showed absent efficacy or contradictory clinical outcomes compared to the no-antiviral treatment group. Nonetheless, there are some drugs that are encouraging. For example, the small molecule oral drugs PAXLOVID and Molnupiravir have been recommended clinically because they significantly reduce the risk of hospitalization and incidence of severe illness or death following early use [184,185,186]. Another small molecule oral drug Azvudine, approved in China, has presented significant efficiency for COVID-19, although its antiviral mechanism is still unclear [187]. In addition, traditional Chinese medicine is believed to have played an important role in COVID-19 pandemic control in China. Three Chinese patent drugs and three formulas for prescriptions with Chinese herbals were recommended to use for COVID-19 after extensive clinical practice [188,189]. The three Chinese patent drugs include Jinhua Qinggan Granule, Lianhua Qingwen Capsule, and Xuebijing Injection. The three formulas include Qingfeipaidu Decoction, Huashibaidu Decoction, and Xuanfeibaidu Decoction. Overall, however, those small-molecule oral drugs or traditional Chinese herbals that are clinically recommended are old drugs for new use, and their efficacies should be further evaluated in clinics. The development of new small-molecule antivirals often takes a long time. Furthermore, with the use of related drugs, it is of concern whether drug-resistant strains will emerge one after another, as in the case of influenza viruses.

## 5. Summarization and Prospects

Based on the epidemiological history of influenza and the current status of the COVID-19 pandemic, both influenza and COVID-19 are likely to have a lasting influence on human health in the future, and the prevention and control of the two diseases will likely be a protracted and lengthy effort.

Since influenza viruses are seasonally prevalent each year, and since the highly variable nature of influenza viruses results in poor time-limited protection with influenza vaccines and limited cross-protection between different subtypes of influenza viruses, an influenza vaccine that can induce the production of broad-spectrum and durable protective antibodies has become a focus of vaccine research in the field. SARS-CoV-2, which has been circulating globally for more than three years, has taken its place in human society, and will possibly coexist with us persistently [190]. However, similarly to influenza vaccines, the currently available SARS-CoV-2 vaccines cannot provide enough protection to at risk populations because of viral mutations or the inducement of non-persistent protective antibodies. The development of a universal vaccine which can induce the production of broad-spectrum protective antibodies is urged in the field. How to extend the protective duration of the vaccine will continue to be a difficult issue in future. As with vaccines, the clinical efficacy of antiviral drugs against the two viruses is still far below people’s expectations. In addition, there are still many unclear aspects in the pathogenesis of influenza and SARS-CoV-2, especially in the mechanism of tissue damage repair. To reduce the occurrence and/or impact of pandemics, it is necessary to conduct research on the pathogenic characteristics, pathogenesis, vaccines and antiviral drugs in order to improve the prevention and control of influenza and COVID-19.

## Figures and Tables

**Figure 1 ijms-24-06369-f001:**
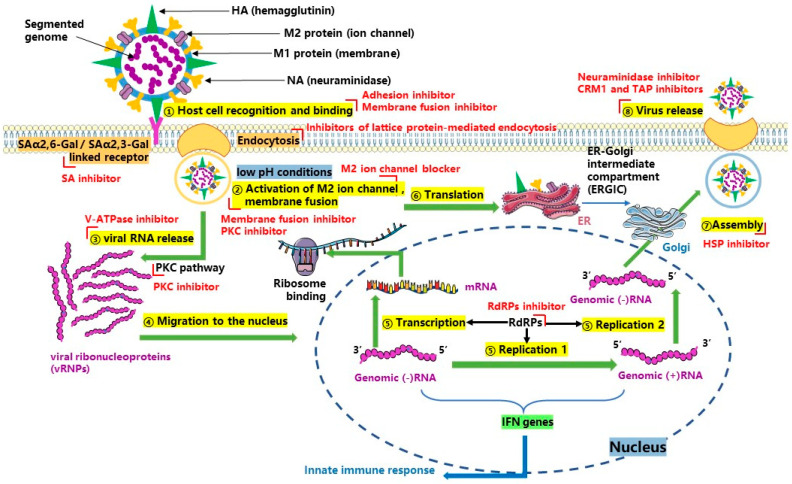
Replication cycle of the influenza virus in host cells, and targets for antiviral drugs and compounds which are marked with the red “∟” symbols.

**Figure 2 ijms-24-06369-f002:**
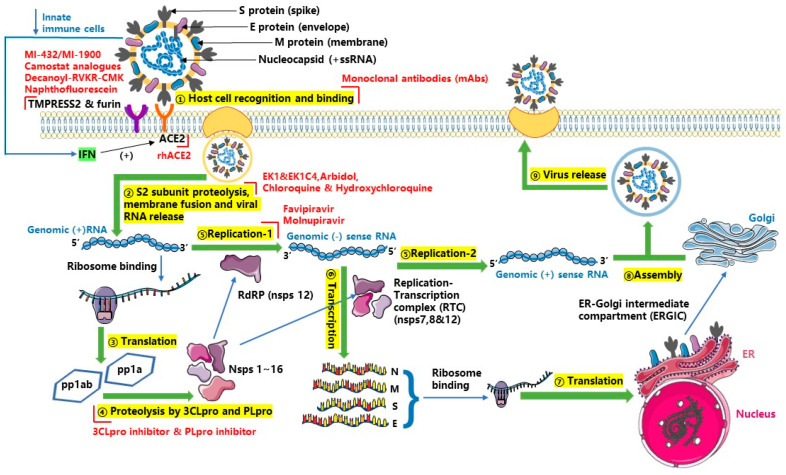
Replication cycle of SARS-CoV-2 in host cells, and targets for antiviral drugs and compounds which are marked with red “∟” symbols.

**Table 1 ijms-24-06369-t001:** Comparison of the basic biological characteristics of influenza virus and SARS-CoV-2.

Characteristics	Influenza Virus	SARS-CoV-2
Genomic characterization	Segmented single-negative strand RNA viruses [2]	Single-stranded positive-strand RNA viruses [3]
Genome size	13.6 kb	29.9 kb [3,4]
Encoded protein	RNA-dependent RNA polymerases (PB1, PB2 and PA), glycoprotein haemagglutinin (HA), glycoprotein neuraminidase (NA), nucleoprotein (NP), matrix protein (M1), membrane protein (M2), nonstructural protein (NS1) and nuclear export protein (NEP/NS2)	Four structural proteins (spike, envelope, membrane and nucleocapsid proteins), 16 nonstructural proteins, and 5–8 accessory proteins have been identified [4,5]
Binding receptor	Sialic acid α2,6/α2,3-galactose (SAα2,6-Gal/SAα2,3-Gal) linked receptors [6]	Human angiotensin converting enzyme 2 (ACE2) [3,7,8]
Morphology of viral particles	Mostly round or slightly ovoid in shape with a diameter of 80–120 nm and a capsule membrane, some in filamentous forms	Mostly spherical or round in shape with a diameter of 60–140 nm in (average size 70–80 nm) and a cobbled surface structure having envelope projections that average 15 ± 2 nm in size
Route of infection	Respiratory transmission, contact transmission	Respiratory transmission, contact transmission
Mode of transmission	Droplets, aerosols	Droplets, aerosols [9]
Transmissibility	High (1918 H1N1 R0: 1.2–1.7, H1N1pdm09 R0: 1.2–2.3) [10,11,12]	High (R0: 2.5~20) [9]

**Table 2 ijms-24-06369-t002:** Representative variants of SARS-CoV-2.

WHO Nomenclature	The GISAID Evolutionary Branch	PANGO Genetic Lineage	First Reported Location	Earliest Reporting Time
Alpha	GR/501Y.V2	B.1.1.7	Britain	September 2020
Beta	GH/501Y.V2	B.1.351	South Africa	May 2020
Gamma	GR/501Y.V3	P.1	Brazil	November 2020
Delta	G/478K.V1	B.1.617.2	India	October 2020
Lambda	GR/452Q.V1	C.37	Peru	August 2020
Omicron	GR/484A	B.1.1.529	South Africa	November 2021

**Table 4 ijms-24-06369-t004:** Characteristics of different COVID-19 vaccines.

Type of Vaccine	Representative Vaccine	Vaccine Effectiveness (VE)
mRNA vaccine	BNT162b2 (Pfizer-BioNTech)	Variable VE from 64% to 95% to SARS-CoV-2 variants within 2–4 weeks of dose two, declining to 45.7% (95% CI, 44.7–46.7%) to Omicron at 10 or more weeks [125,126,127].
	mRNA-1273 (Moderna)	Variable VE from 75.15 to 96.4% to SARS-CoV-2 variants within 2–4 weeks of dose two; 44.0% (95% CI, 35.1–51.6%) against Omicron infection at 14–90 days but declining quickly [125,128,129].
Viral vector vaccine	AZD1222 (AstraZeneca–University of Oxford)	Variable VE from 10% to 81% to SARS-CoV-2 variants [130,131]; No effect against the Omicron variant from 20 weeks [125].
	Ad26.COV2-S (Johnson & Johnson)	Variable VE from 57% to 70% to SARS-CoV-2 variants [132,133]; A single dose provides 52.9% protection against moderate to severe-critical COVID-19 [134].
Inactivated virus vaccine	CoronaVac (Sinovac)	Variable VE from 48.3% to 90% to SARS-CoV-2 variants [135,136].
Subunit vaccine	NVX-CoV2373 (Novavax)	Variable VE from 43% to 89% to SARS-CoV-2 variants [137,138].

**Table 5 ijms-24-06369-t005:** Characteristics of anti-influenza virus drugs or compounds.

Type	Name of the Drug or Compound	Route	Clinical Effectiveness	Notes
**Drugs or Compounds that Act on Viral Replication**				
Neuraminidase inhibitors (NAIs)	Zanamivir (trade name: Relenza)	Taken orally in pill or liquid	The duration of the disease can be reduced to 1–2.5 days in patients who receive treatment within 48 h of symptom onset. Administered twice daily [144]. IC50 = 0.24 ± 0.02 nM (for H1N1) [145].	Emerged virus strain with drug resistance or declined sensitivity [146].
	Oseltamivir (trade name: Tamiflu)	Inhalation/intravenous injection	In the intention-to-treat infected population, the time to remission of all symptoms was 21% shorter compared to placebo recipients (time ratio 0.79, 95% CI 0.74–0.85; *p* < 0.0001). Administered twice daily [147]. IC50 = 0.07 nM (for H1N1) [145].	Approximately 0.5–1.0% of community A(H1N1) pdm09 isolates are currently resistant to oseltamivir [148].
	Peramivir (trade name: Rapiacta, Perami flu, Rapivab)	Intravenous injection	Median time to alleviation of symptoms was 29.1 h. No difference between peramivir and oseltamivir groups for time to defervescence and viral titers declined. Peramivir only requires a single daily dose [149].	Low oral bioavailability.
	Laninamivir	Inhalation	The difference between the 40 mg laninamivir octanoate and oseltamivir groups was not statistically significant. Laninamivir only requires a single daily dose [150].	Lanamivir is structurally modified zanamivir.
	AV5080	Oral	IC50 = 0.06 nM(for H1N1) [145].	Highly active against oseltamivir-resistant influenza viruses.
M2 ion channel blockers	Amantadine (trade name: Gocovri)	Taken orally in capsule/tablet, syrup	Severe drug resistance to circulating influenza virus.	Not recommended for use in clinics.
	Rimantadine (trade name: Flumadine)	Taken orally in capsule/tablet, syrup	Severe drug resistance to circulating influenza virus.	Not recommended for use in clinics.
	Diazabicyclooctane (DABCO)	Not yet available as a drug	-	The dual effect of spatial and electrostatic potential barriers
RdRP inhibitors: PB inhibitors	Ribavirin	Inhalation/intravenous injection/tablet	Combination of oral amantadine, ribavirin, and oseltamivir (TCAD) was associated with significantly greater antiviral effects than oseltamivir monotherapy (40.0% of TCAD versus 50.0% of oseltamivir recipients had detectable viral RNA on day 3) [151].	Inconsistent in vitro and in vivo efficacy.
	Pimodivir	Oral	EC50 = 0.6 nM (for H3N2) [151]. In a mouse model, 100% survival was achieved at 96 h post-infection administration [152].	Delayed administration showed rapid inhibition of mRNA production.
RdRP inhibitors: nucleoside analogues	Favipiravir	Oral	Favipiravir demonstrated significantly faster time to alleviation of influenza symptoms (median, 82.3 versus 97.3 h) and viral load reductions compared with the placebo group. EC50s: 0.03–0.94 μg/mL (those resistant to adamantanes and NAIs), 0.09–0.83 μg/mL(H5N1), and 0.06–3.53 μg/mL(H7N9) [151].	Possible adverse reactions.
RdRP inhibitors: cap-dependent nucleic acid endonuclease inhibitor	Baloxavir marboxil	Oral	Median time to symptom improvement was similar (53.5 vs. 53.8 h) between the baloxavir and oseltamivir groups. Duration of detectable virus was shorter in the baloxavir group compared to oseltamivir (24 vs. 72 h, *p* < 0.001) [153]. EC50s: 0.63–0.77 ng/mL (H1N1), 0.36–0.68 ng/mL(H3N2), and 2.7–4.1 ng/mL(IFV B) [151].	High plasma retention time.
Hemagglutinin inhibitors: adhesion inhibitors	LPG10SA	Not yet available as a drug	-	Sialic acid glycoside with a dendritic polyglycerol backbone.
	2-Deoxyuridine analogue	Not yet available as a drug	-	No cytotoxic effects at therapeutic doses.
	Neoechinulin B	Not yet available as a drug	-	Natural product.
Hemagglutinin inhibitors: membrane fusion inhibitors	Arbidol	Oral	IC50s:12 μΜ(H3N2), 13.3 μΜ(IFV B) [154].	Dual interactions with membranes and aromatic amino acids in proteins.
	MBX2329, MBX256	Not yet available as a drug	-	-
Nucleoprotein inhibitors	Nucleozin	Not yet available as a drug	-	-
**Drugs or Compounds that Act on the Host**				
SA inhibitor	DAS181	Inhalation	1 mg/kg/day given prophylactically was found to protect against H5N1 infection, and 0.7 mg/kg/day reduced mortality by 94% [155].	Recombinant sialidase fusion protein.
Type II transmembrane serine proteases (TTSPs) inhibitors	Camostat	Intravenous injection/tablet	IC50/EC50 = 4.4 μM [156].	Inhibit its cleavage of the precursor protein HA0.
	Bromohexine hydrochloride	Not yet available as a drug	-	-
	Peptidomimetic inhibitors	Not yet available as a drug	-	-
Inhibitors of lattice protein-mediated endocytosis	Pitstops 1 and 2	Not yet available as a drug	-	Inhibits lattice-protein-mediated endocytosis to block viral invasion with no significant effect on cell viability.
	Chlorpromazine	Intravenous injection/oral	No available data	Acts downstream on lattice-protein encapsulated vesicle formation.
Vesicular proton ATPase(V-ATPase) inhibitors	Bafilomycin	Not yet available as a drug	-	Poor water solubility and potential off-target effect
	diphyllin and bafilomycin nanoparticles	Not yet available as a drug	-	Uses PEG-PLGA diblock copolymers. Lower cytotoxicity and greater in vitro antiviral activity.
Protein kinase C (PKC) inhibitor	Bisindolylmaleimide (GF109203X)	Not yet available as a drug	-	A highly specific PKC inhibitor that is active against most PKC isoforms.
Heat shock protein (HSP) inhibitor	Geldanamycin	Not yet available as a drug	-	Its binding to the N-terminal adenosine triphosphate binding pocket of Hsp90 inhibits the function of the protein chaperone.
CRM1 and TAP inhibitors	5,6-dichloro-1-β-D-ribofuanosyl-H-benzimidazole (DRB)	Not yet available as a drug	-	Inhibition of nuclear transport in the CRM1 and TAP-P15 pathways, completely blocking influenza virus export, however, LB is toxic in the body.
RK pathway inhibitor	CI-1040	Not yet available as a drug	-	A non-competitive MEK1/2 inhibitor of adenosine triphosphate.

## Data Availability

All data in this paper are from published papers. The contents of this article are solely the responsibility of the authors and do not necessarily represent the views of China CDC and other organizations.

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
