# Peer review of "A Comparison of Etiology, Pathogenesis, Vaccinal and Antiviral Drug Development between Influenza and COVID-19"

_ijms, 2023, doi:10.3390/ijms24076369_

Round 1
Reviewer 1 Report
In this review, the authors gathered the information related to etiology, pathogenesis, vaccinal and antiviral drug development of influenza and COVID-19.
This is a timely and important review. However, I felt that the review lacks a more relevant synthesis and insightful information. The authors should clearly state what similarities or differences are there in these two viruses. Some questions and concerns are listed below.
Major points:
1. When talking about pathogenesis, usually we will discuss two aspects, including the virus itself and the host immune response. It would be better that the authors could provide more in-depth information about the mechanism of pathogenicity.
2. As to the vaccine topic, it is widely recognized that RNA viruses undergo mutation, which affects the efficacy of vaccine. The authors should address the mutation mechanisms of influenza and SARS-CoV-2. Do they use the same or different mechanisms of mutation? (Such as point mutation, reassortment, recombination, etc.) What new insights will they bring?
Minor points:
1. Table1: why the encoded protein for influenza NS1 is annotated with “not shown”? Besides, the binding receptor for influenza virus is usually demonstrated as “sialic acid α2,6/α2,3-galactose (SAα2,6-Gal / SAα2,3-Gal) linked receptors”.
2. Table1: The full name for ACE2 is “angiotensin converting enzyme 2”.
3. Line 240: there is a typo here. It is supposed to be “IFN-a/b in infected …”.
4. Line 384: there is a grammatical error here. … bind the sialic acid (SA) receptor to “adsorb” to the cell…
5. Could SARS-CoV-2, like the flu, transmit from animals to humans?
6. Will co-infection of influenza influence the pathogenicity of COVID? Please describe the issue about co-infection of SARS-CoV-2 and influenza virus.
Reviewer 2 Report
The review is aimed at reviewing the literature in etiology, pathogenesis, vaccine and antiviral strategies of influenza and COVID19. Overall, the review is complete, but some points of the text should be better discussed and more recent literature should be added:
- In the pathogenetic mechanisms of influenza and SARS CoV-2 infections (sections 2.3.1 and 2.3.2), the most recent evidence from literature and some mechanisms involved in viral pathogenesis should be better discussed. For instance, several authors have reported the role of intracellular redox state modulation as a potential pathogenetic mechanism in both viral infections. Others have identified several redox factors modulated during infections that are also involved in the regulation of the inflammatory response. This aspect should be described more in detail.
- In antiviral drugs paragraph, the authors summarize the main treatment of influenza and SARS-CoV-2 viral infections targeting specific viral structures. Overall, this section could be deepened and improved by adding new alternative therapeutic strategies which are currently being studied in both models of infections. In different works it has been proposed natural or synthetic molecules able to damage viral structure, the binding with host receptors or the late stage of viral replicative cycle. A figure representing influenza virus replicative cycle and the main anti-viral targets could also be added.
- Some typing mistakes are present, revise carefully the text.
Round 2
Reviewer 1 Report
The revised MS entitled “The comparison of etiology, pathogenesis, vaccinal and antiviral drug development between influenza and COVID-19” has relieved my concerns and is acceptable now.